# COVID-19 and Changes in the Model of Physical Fitness and Body Composition of Young Tennis Players

Rok Blagus [1,2], Vedran Hadzic [3], Angel Ivan Fernandez Garcia [4,5], Bojan Leskosek [3], Benjamin J. Narang [3,6] and Ales Filipcic [3,*]

1    Institute for Biostatistics and Medical Informatics, Faculty of Medicine, University of Ljubljana, 1000 Ljubljana, Slovenia; rok.blagus@mf.uni-lj.si
2    Faculty of Mathematics, Natural Sciences and Information Technologies, University of Primorska, 6000 Koper, Slovenia
3    Faculty of Sport, University of Ljubljana, 1000 Ljubljana, Slovenia; vedran.hadzic@fsp.uni-lj.si (V.H.); bojan.leskosek@fsp.uni-lj.si (B.L.); benjamin.narang@ijs.si (B.J.N.)
4    Growth, Exercise, Nutrition and Development (GENUD) Research Group, University of Zaragoza, 50009 Zaragoza, Spain; angelivanfg@unizar.es
5    Faculty of Health and Sport Sciences, Department of Physiatry and Nursing, University of Zaragoza, 50009 Huesca, Spain
6    Department of Automatics, Biocybernetics and Robotics, Jožef Stefan Institute, 1000 Ljubljana, Slovenia
*    Correspondence: ales.filipcic@fsp.uni-lj.si; Tel.: +386-4-170-48-76

**Abstract:** This retrospective study aimed to estimate the differences between selected indicators of physical fitness and body composition in young tennis players during the COVID-19 pandemic (2020 and 2021) and their values as predicted using the pre-pandemic trend (2015–2019). Data were collected from selected boys (mean ± SD; 13.2 ± 1.7 years) and girls (13.1 ± 1.9 years) during annual tests. Data were analyzed with linear mixed-effects models for males and females, separately, to predict body composition and physical fitness test scores, adjusting for age and pre-pandemic trends in the data. Compared with expected values, body fat mass increased in boys (2020: 0.68; 0.44–0.92, 2021: 1.08; 0.72–1.43), whereas muscle mass decreased (2020: −0.22; −0.34−−0.10, 2021: −0.28; −0.46−−0.10) throughout the pandemic. Interestingly, boys' age-adjusted squat jump test scores improved relative to their expected scores during COVID-19 (2020: 0.19; 0.00–0.38, 2021: 0.35; 0.06–0.63). No other differences between predicted and measured values were noted across the observation period. The results of this study suggest that the sustained reduction in sports activity caused by the pandemic may have negatively affected the body composition of athletes; however, this did not affect selected performance indicators.

**Keywords:** COVID-19; tennis; body mass; physical fitness; performance; testing

## 1. Introduction

COVID-19, caused by the SARS-CoV-2 virus, is a multisystem disease that can affect the pulmonary and cardiovascular systems. The virus affects the ability of many people to exercise and benefit from sports participation [1]. In particular, the Slovenian Government implemented a strict first lockdown from 12 March 2020 to 15 May 2020 as a preventive measure to limit the spread of the disease. The second lockdown was introduced during the second wave of COVID-19, from 18 October 2020 to 15 May 2021. All schools were required to close during both lockdown periods; therefore, lessons were carried out online.

The strict measures adopted to stabilize the healthcare system had some direct and some indirect societal consequences, particularly for young athletes. Specifically, youth athletes are characterized by developing intrinsic dual careers (i.e., a combination of academic and athletic activities), and, therefore, simultaneously face several athletic, academic, psychological, psychosocial, financial, and legal challenges [2]. These individuals were strongly affected by COVID-19 due to numerous factors, including new academic realities,

the loss of practice and competition, and social distance from teammates. Isolation during the pandemic led to radical changes in athletes' lifestyles, which affected both their physical activity levels and dietary habits. [3]. The recurring, widespread suspensions (travel restrictions, social distancing, and closure of training facilities) also prevented athletes from continuing their usual training programmes [4], and competitions were suspended at all competitive levels.

While each sport was affected differently, depending on its particular characteristics, the global outbreak of COVID-19 certainly had a considerable impact on tennis [5]. Importantly, tennis is a highly demanding sport characterized by repeated high-intensity efforts during a variable match time. This requires highly competitive players to excel across a range of different fitness components, such as power, speed, agility, and endurance [6,7]. These components are influenced by body composition parameters [8–10], which are often used to evaluate an athlete's potential to compete at the highest level [11,12].

The observed changes in physical activity and dietary habits induced by COVID-19 may have influenced the body composition of tennis players. Physical activity and dietary patterns are known to change during extended breaks (e.g., summer breaks, off-season, injuries, etc.), especially if they occur unexpectedly, as in the case of COVID-19 lockdowns [13]. Although attempts were made to compensate for these changes with alternative strategies such as training at home [14,15], it is important to assess whether these strategies provided sufficient stimuli for competitive athletes [16] to prevent negative changes in body composition. Many studies that have examined the effects of COVID-19 restrictions on physical fitness [17–20] or body composition have been conducted on professional athletes and have shown conflicting results [21–23]. As body composition fluctuates and develops in young athletes on a regular basis, it is difficult to identify the extent to which changes can be attributed to the effects of COVID-19. The aim of this retrospective study was, therefore, to estimate the differences in the selected physical fitness and body composition indicators during the COVID-19 pandemic (2020 and 2021) and to compare these values to those predicted using the pre-COVID-19 trend (2015–2019).

## 2. Materials and Methods

### 2.1. Research Design and Participants

The data were collected as part of the annual testing programme of selected young tennis players aged 10 to 18 years. The programme is organized by the National Tennis Federation and conducted by the Institute of Sport every year in the second week of October. The participants were selected by the coach of the national team and included in the list of junior national teams, which means that they were among the top ten tennis players, nationally, in their age group. During the annual testing, data were collected from the young tennis players on various aspects of motor- and tennis-specific skills and body composition. Written informed consent was obtained from all athletes and their legal guardians, permitting the testing procedures to be performed and anonymised data to be collected, analysed, and archived. The study was approved by the Ethics Committee of the University of Ljubljana, Faculty of Sport (2-2023).

The following number of athletes were included within each: 61 boys and 43 girls in 2015, 70 boys and 48 girls in 2016, 60 boys and 50 girls in 2017, 66 boys and 43 girls in 2018, 63 boys and 43 girls in 2019, 30 boys and 17 girls in 2020, and 52 boys and 37 girls in 2021. The average age of boys was 13.2 ± 1.7 years, and that of girls was 13.1 ± 1.9 years.

### 2.2. Data Collection

The testing protocol was identical each year. After arrival, registration, and a 15-min supervised warm-up, the testing and screening procedures were performed in the following order: (1) anthropometric measurements, analysis of body composition; (2) postural analysis and functional movement screening; (3) speed, agility, flexibility, coordination and power tests, repetitive strength, and dynamometric strength measurements; (4) force plate

tests; (5) test of aerobic endurance; (6) a questionnaire about training, competition results, and injuries.

Based on theoretical considerations, and to investigate different aspects of body composition and physical fitness that are particularly relevant to tennis, six variables were included in the data analysis. Body fat mass (BFAT) and body muscle mass (BMUSC) (kg) were measured using the InBody 720 Octopolar Bioimpedance device (Biospace, Seoul, Republic of Korea). Squat jump (SJ) and countermovement jump (CMJ) height (cm) were measured on a force plate (Kistler, 9287, Winterthur, Switzerland). Upper body explosive power was measured using the overhead medicine ball throw (OMBT), and $\dot{V}O_2$ max (mL/kg/min) was estimated using 20-m progressive shuttle runs (BEEPVO2M) to exhaustion [24].

In 2020 and 2021, the annual testing of tennis players was conducted 5 months after the end of the first lockdown (Figure 1). During these 5 summer months, the young tennis players trained at a normal level (as before the COVID-19 pandemic) and participated in national and international junior tournaments.

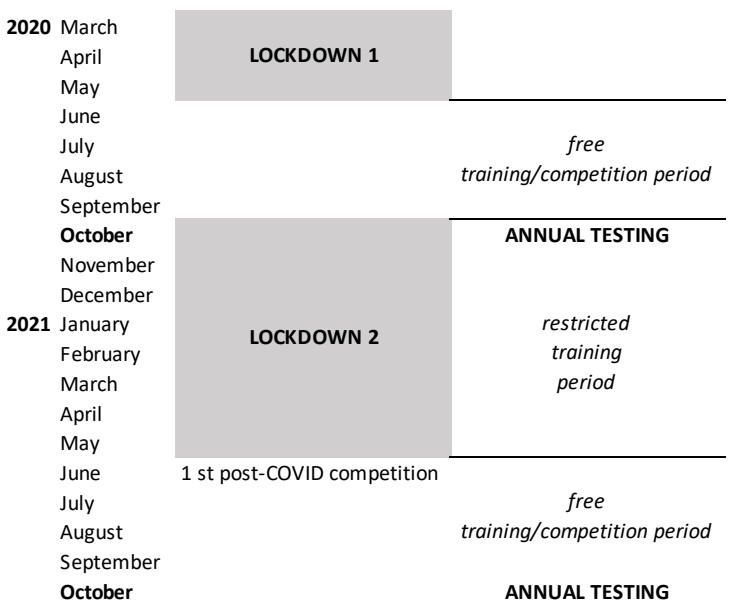

**Figure 1.** Overview of COVID-19 lockdowns and training/competition schedule for young tennis players.

### 2.3. Statistical Analysis

The data were analysed using linear mixed-effects models (LMEs), separately for males and females, with the random intercept and random slope for age by subject ID included in the model. For each of the six outcomes (BEEPVO2M, BFAT, BMUSC, CMJ, OMBT, SJ), the fixed effects part of the model contained age (modelled as a non-linear association using cubic splines), period (2015–2019, 2020, 2021) as a three-level factor, and year difference (set to 0 for 2019–2021 and to −1, −2, −3, and −4 for 2018, 2017, 2016, and 2015, respectively), which was modelled as a non-linear association using natural splines to allow extrapolation. A square-root transformation was applied to the outcome variable to stabilize the variance. This model specification assures that only the data from 2015 to 2019 were used to estimate the general (age-adjusted) trend. This trend was then used to make predictions for the 2020–2021 period. The number and the position of the knots when constructing the splines were set in such a way that the fit of the model was adequate. Correct specification of the two design matrices (i.e., the goodness-of-fit (GoF) analysis) was verified as proposed by Peterlin [25]: the considered subset F processes included the age (including its respective nonlinear terms) and year difference (including its respective nonlinear terms); note that a significant *p*-value indicates an inadequate model fit. It was specified a priori that only

two contrasts will be considered: the difference between the age-adjusted effect in 2020 (and 2021) and the age-adjusted predicted effect in 2020 (and 2021). Positive values of the contrast suggest that the age-adjusted outcome in a particular year (2020 or 2021) was larger than it ought to have been given the age-adjusted, pre-COVID trend; negative values of the contrast suggest that the value was smaller. The family-wise-error rate (FWER) was controlled using the procedure proposed by Hothorn [26]. Appropriate effect size measures (e.g., Cohen's *d*) were computed (Supplementary Material). A sensitivity analysis was also performed, in which the number of knots in the natural spline for the year difference was varied. Given that these sensitivity analyses did not affect the ultimate inferences, these results are not reported.

An exploratory power analysis was conducted by performing a Monte Carlo simulation study. We assumed that the trend is given by using the following equation: $y = 5 + 0.1x - 0.01x^2$, $x \in \{0, 1, \ldots, 6\}$, with a normally distributed random error around this trend with a mean of 0 and a standard deviation of 0.5. A study with 300 independent observations pre-COVID (uniformly distributed over the interval $x \in \{0, 1, \ldots, 4\}$) and 30 and 50 observations one and two years after COVID ($x \in \{5\}$ and $x \in \{6\}$), respectively, would have a 95% power to detect a 10% decline from the estimated trend (using only pre-COVID data) one year after COVID ($x \in \{5\}$). A study with 225 independent observations pre-COVID (uniformly distributed over the interval $x \in \{0, 1, \ldots, 4\}$) and 20 and 40 observations one and two years after COVID ($x \in \{5\}$ and $x \in \{6\}$), respectively, would have a 85% power to detect a 10% change. Note that the first and second sample sizes are approximately equal to the sample sizes obtained in our study for males and females, respectively. Also, note that the power calculation is conservative since independent observations are assumed (in practice, the power is expected to be larger as we can reasonably assume that any two observations from the same individual are positively correlated).

The analysis was performed using R (R version 3.6.3.) [27]. LMEs were fitted using the R package [28]. Basis matrices for splines were generated using the package **splines**. The GoF analysis was performed using the package **gofLMM**. The contrasts were estimated using the package **multcomp**. The effect size measures were computed using the package **effectsize** [29]. All the reported 95% confidence intervals (CIs) and *p*-values are two-sided. Estimates and CIs are reported in the text as (Estimate; 95% CI range). A *p*-value of less than 0.05 was considered statistically significant.

## 3. Results

Figure 2 contains the descriptive data, including medians, interquartile ranges, and the number of measurements (*n*) for age and for each outcome, split by gender and year. The estimated contrasts obtained from the fitted models are summarized in Figure 3. Supplementary Tables S1–S7 report the summaries of the fitted models, including the results of the GoF analyses and the estimated effect sizes. Significantly larger age-adjusted values of BFAT were observed for males in 2020 (0.68; 0.44–0.92) and 2021 (1.08; 0.72–1.43) than what would be predicted using the age-adjusted, pre-COVID trend. Similarly, the age-adjusted values of SJ observed for males in 2020 (0.19; 0.00–0.38) and 2021 (0.35; 0.06–0.63) were also significantly larger than the pre-COVID trend model would have predicted. Smaller age-adjusted values of BMUSC were also observed for males in both 2020 ($-0.22$; $-0.34$–$-0.10$) and 2021 ($-0.28$; $-0.46$–$-0.10$) than predicted by the age-adjusted, pre-COVID trend. In male athletes, there were deviations from the trend in the period before COVID-19 for BFAT and BMUSC (2020: *p* = 0.01; 2021: *p* = 0.01). There were no other significant differences between the observed and predicted age-adjusted values (Figure 3; see also Supplementary Materials).

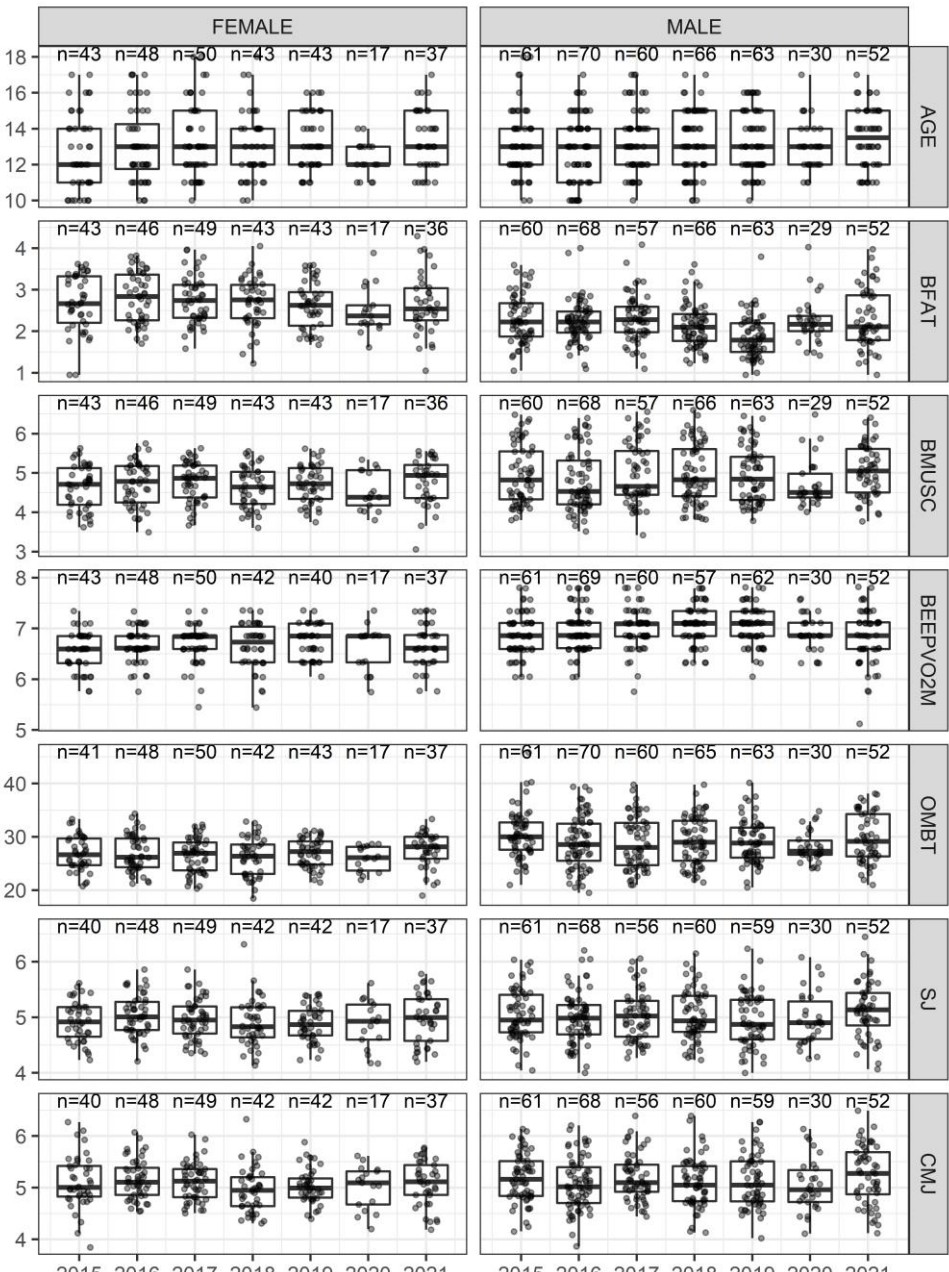

**Figure 2.** Boxplots for age and the six main (square-root transformed) outcomes (different rows), with respect to gender (two columns) and year (*x*-axes). Note: Individual data points are superimposed over descriptive statistics. Data points are jittered to prevent overplotting. AGE—athlete chronological age; BFAT—body fat mass; BMUSC—body muscle mass; OMBT—overhead medicine ball throw; BEEPVO2M—20 m shuttle run test; CJM—countermovement jump height; SJ—Squat jump height.

In female athletes, there were no deviations from the trend in the period before COVID-19 for body composition (BFAT, BMUSC) and physical performance parameters (Figure 3). For aerobic endurance in the 20-m shuttle run test, the results were worse than before COVID-19, especially in 2021, but not significantly (2020: *p* = 0.27; 2021: *p* = 0.07). For physical performance parameters, the only significant change observed was an improvement in SJ height for males. There was also a non-significant positive trend in CMJ values (2020: *p* = 0.5; 2021: *p* = 0.37). All other parameters were not significantly different from the expected model.

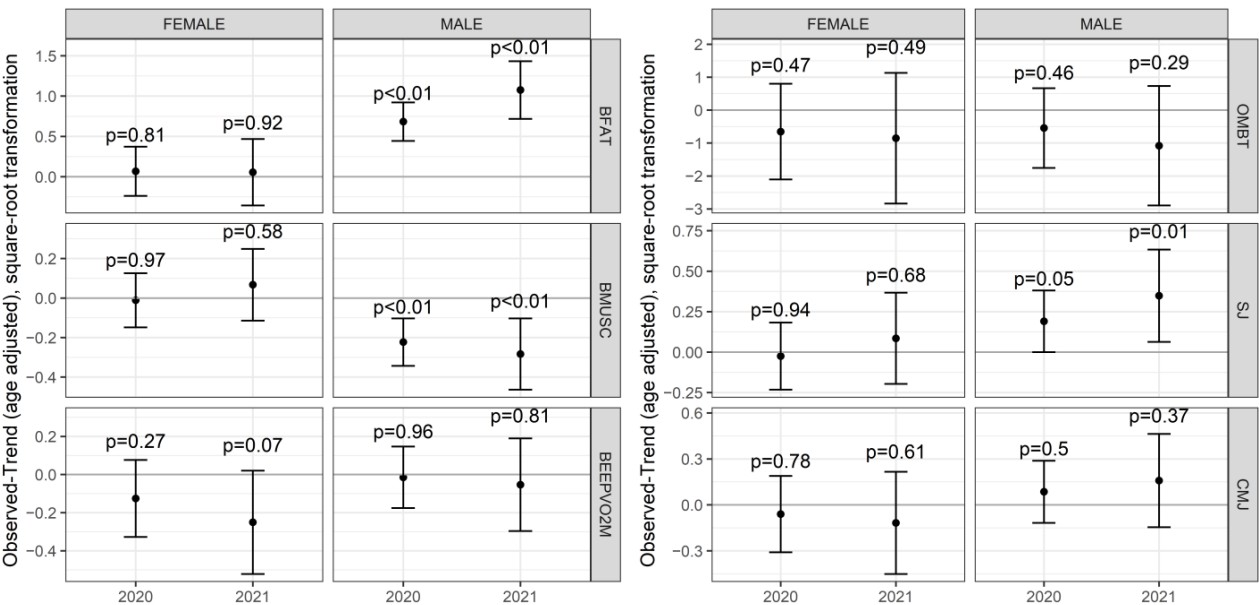

**Figure 3.** The difference between the age-adjusted effect in 2020 and 2021 and the age-adjusted predicted effect in 2020 and 2021 by sex with corresponding 95% confidence intervals and *p*-values. Note: The *y*-axis shows the estimated (age-adjusted) difference between the observed and predicted effect based on the estimated pre-COVID, non-linear trend, applying the square-root transformation to outcomes. Positive (negative) values of the contrast suggest that the age-adjusted outcome in a particular year (2020 or 2021) was larger (smaller) than that predicted by the age-adjusted, pre-COVID, non-linear trend. BFAT—body fat mass; BMUSC—body muscle mass; OMBT—overhead medicine ball throw; BEEPVO2M—20 m shuttle run test; CJM—countermovement jump height; SJ—Squat jump height.

## 4. Discussion

This study investigated the differences between selected physical fitness and body composition indicators during the COVID-19 pandemic (2020 and 2021) and their values as predicted using the pre-COVID-19 trend (2015–2019). Historical data from 227 female and 320 male tennis players collected before the pandemic were used to predict the expected age-adjusted evolution of body composition and physical fitness variables. This prediction was then compared to the age-adjusted measurements recorded during the COVID-19 pandemic. To our knowledge, this is the first study to analyse changes in body composition and physical fitness in young tennis players using this approach.

In young male athletes, we observed significant changes in body composition relative to predicted values, which was not the case in female athletes. Specifically, body fat mass scores were significantly higher, and body muscle mass scores significantly lower, than the expected results based on the pre-COVID trend. These negative changes in body composition in boys persisted after the first lockdown and worsened after the second lockdown.

The significant differences in body composition that we observed in young male tennis players as a result of the COVID-19 pandemic may indicate the negative effects of increased sedentary behaviour, as was also observed in a study of young badminton players in the COVID-19 period [30]. A significant increase in fat mass and a decrease in lean mass (measured by DXA) were also observed in male American football players, despite the fact that they reported strength training as their most frequent activity during the suspension period [21]. However, in a study by Yasuda et al. [22], in which the body composition of fencing athletes was measured at three different time points (September 2019, June 2020, and September 2020), the authors reported an increase in fat mass only in women, which normalised at the third measurement. In another study [13], the body weight, fat mass, and muscle mass of university athletes were analysed using bioelectrical

impedance before suspension (January 2020) and shortly after the resumption of on-campus training (August/September 2020). A decrease in fat mass was observed in men, whereas fat mass increased in women. Spyrou et al. [17] found no difference in body composition in elite futsal players when assessed twice across a three-month period using skinfold measurements. Body fat and muscle mass also remained unchanged in a group of female soccer players between August 2020 and February 2022 [31]. In our study, we also found no changes in body fat mass and muscle mass in female tennis players. This could be explained by behaviours relating to excessive concerns with weight and body image, as has been reported in young female tennis players [32]. We can also assume that the changes in body composition and physical fitness that occurred during the period of isolation disappeared or decreased by the time testing was conducted.

For physical performance parameters, the only significant change observed in our study was an improvement in SJ height and a non-significant positive trend in CMJ values. Obayashi et al. [33] found non-significant differences in the body composition of adolescent athletes in three team sports and two individual sports from August 2019 to August 2020 using bioimpedance. Moreover, Campa et al. [34] compared the body composition results of Serie A soccer players collected using bioelectrical impedance in the COVID-19 season and in the regular season and concluded that body fat mass remained unchanged while muscle mass and the phase angle of muscle mass decreased in the COVID-19 season. In the regular season, body fat and muscle mass did not change, while the phase angle of muscle mass increased.

Since the young tennis players assessed in this study trained normally in the period between the two measurements, the causes of the negative changes in body composition in males could be attributed to dietary habits and/or long periods of online schooling. This led to a decline in physical fitness in a population of 20,000 schoolchildren while isolation measures were in place. The largest decline in performance was in aerobic endurance, followed by a decline in whole-body coordination. The smallest decline was in explosive strength, although the results were still worse than in previous years [35].

In summary, although it is intuitive to expect increases in body fat mass and decreases in muscle mass due to lower levels of physical activity during COVID-19 lockdowns, it seems that changes reported in the literature are rather random regardless of the method used to assess body composition. There have been studies where body composition changes were observed and were sex-specific, while, in other studies, the changes were non-significant without sex differences. An important novelty of this study is the comparison of observed differences against those predicted based on the changes in body composition of the same participants during the pre-COVID period. In adopting this approach, the general changes in the body composition and physical fitness of athletes that may have already been occurring prior to the COVID-19 pandemic could be accounted for.

In addition to analysing changes in body composition, many studies have also examined changes in physical fitness. In the present study, the performance of male and female tennis players in the physical fitness tests compared to the time before COVID-19 was mostly not significant, except for significantly higher squat jump test scores, specifically in boys.

The observed increase in squat jump height in the male participants was an unexpected finding. The improvement was significantly greater in 2021 compared to before the pandemic. This is not consistent with a study examining the effects of prolonged training interruption [18] in soccer players, in which a decrease in countermovement jump height was observed without significant changes in squat jump height. A decrease in vertical jump height after COVID-19 has also been observed in other studies [19,36]. Tan et al. [37] reported that athletes with planned post-COVID-19 training required between 2 and 4 weeks to return their jumping capacity to their pre-COVID-19 level. In addition to those studies that observed impairments in squat and countermovement jump performance induced by the pandemic, some studies have found no significant changes associated with the occurrence of COVID-19 [17]. Moreover, in accordance with our findings, there are also studies

showing the positive effects of home training during lockdown on jump performance in soccer players [38] and young tennis players [39]. This could support the increase in jump height observed in our study, as the specific change in training practices adopted by our participants may have induced a training stimulus conducive to increases in lower-limb power. However, we were unable to collect specific training practice data from our participants during quarantine. In addition, the results appear contradictory, as the decrease in muscle mass with a concomitant increase in fat mass should theoretically have reduced SJ performance.

CMJ performance is typically superior to SJ performance, and the difference in performance is thought to be due to the effective use of the stretch-shortening cycle during countermovement [40], better ability to store and use elastic energy [41], and muscle fibre type [42]. We used three common methods to measure the magnification of the stretch-shortening cycle in order to better understand the surprising SJ performance changes: (1) direct comparison (CMJ height—SJ height [cm]); (2) magnification before stretching, ((CMJ − SJ)/SJ) × 100 [%]); (3) ratio of eccentric loading, (CMJ ÷ SJ [ratio]) (Figure 4). In 2020 and 2021, the age-adjusted indices in male athletes were lower than what would be predicted by the age-adjusted, pre-COVID trend, mainly due to the significant increase in SJ results during that period. Under similar circumstances, Paravlic et al. [43] investigated the contractile properties of skeletal muscle in soccer players, observing no deterioration in lower limb power during the COVID-19 period. This could mainly explain the suboptimal conditioning and poorer ability to perform eccentric–concentric leg movements, which was the result of a change in lifestyle during the COVID-19 period and the too-short implementation time of systematic training for the development of all forms of jumping performance.

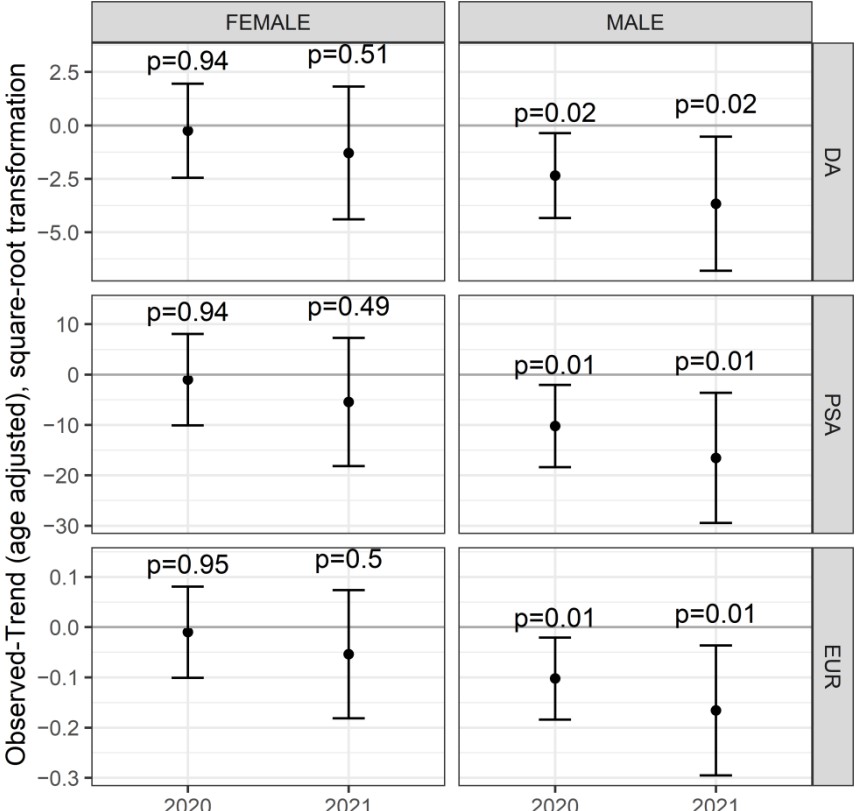

**Figure 4.** The difference between the age-adjusted effect in 2020 and 2021 and the age-adjusted predicted effect in 2020 and 2021 by sex. Positive (negative) values of the contrast suggest that the age-adjusted outcome in a particular year (2020 or 2021) was larger (smaller) than that predicted by the age-adjusted, pre-COVID, non-linear trend. Note: DA = direct comparison; PSA = Pre-stretch Augmentation; EUR = Eccentric Utilization Ratio.

An important limitation of this study is the considerable reduction in sample size for the 2020 measurements (Figure 2). This likely occurred due to concerns from parents and guardians to allow the young athletes to participate in the annual testing programme during the pandemic. We have attempted to account for this dropout in our analysis by stratifying by sex and adjusting for age. However, there could be reasons other than age and sex for dropout that we were unable to account for. It should be noted that the dropout rate was highest in 2020, when only 37% of girls and 47% of boys participated in the tests (the percentage refers to the average number of participants from 2015 to 2019); however, the situation improved significantly in 2021, when 81% of participants of both sexes participated in the annual tests. In most cases, there was no difference in the inference between the 2020 and 2021 comparisons. This consistency perhaps indicates that the results still appear to be valid, which is in line with the principles underlying the sensitivity analyses.

## 5. Conclusions

In conclusion, the present study demonstrates that the COVID-19 restrictions resulted in significant changes in fat and muscle mass in young male tennis players, without a decrease in their physical fitness. Interestingly, we observed an increase in squat jump performance in these young male athletes during the COVID-19 period. In young female tennis players, we found no significant differences in body composition or physical performance during the observation period. The results of this study suggest that planned or unexpected prolonged training interruption may require countermeasures (e.g., structured home training and adjustment of dietary habits) to attenuate the potential for negative changes in body composition. However, it seems that changes in body composition associated with these periods of interrupted training do not necessarily result in clear decreases in the physical fitness test performance of young athletes.

**Supplementary Materials:** The following supporting information can be downloaded at: https://www.mdpi.com/article/10.3390/app131810015/s1. The data are the output of the conducted research, see Supplementary Materials. Supplementary Table S1: The estimated contrasts (Estimate), 95% confidence intervals (CI), and p-values for the differences between the age-adjusted effects in 2020 and 2021 and the age-adjusted predicted effects in 2020 and 2021 by sex and (square-root transformed) outcome. Supplementary Table S2: Estimated coefficients, standard errors (SEs), p-values, partial $eta^2$, and Cohen's d, direct comparison (DC). Supplementary Table S3: Estimated coefficients, standard errors (SE), p-values, partial $eta^2$, and Cohen's d. Pre-stretch Augmentation (PSA). Supplementary Table S4: Estimated coefficients, standard errors (SE), p-values, partial $eta^2$, and Cohen's d. Eccentric Utilization Ratio (EUR). Supplementary Table S5: The estimated contrasts (Estimate), 95% confidence intervals (CI), and p-values for the differences between the age-adjusted effects in 2020 and 2021 and the age-adjusted predicted effects in 2020 and 2021 by sex and (square-root transformed) outcome. Supplementary Table S6: Estimated coefficients, standard errors (SE), p-values, partial $eta^2$, and Cohen's d. OMBT. Supplementary Table S7: Estimated coefficients, standard errors (SE), p-values, partial $eta^2$, and Cohen's d. CMJ. Supplementary Table S8: Estimated coefficients, standard errors (SE), p-values, partial $eta^2$, and Cohen's d. BEEPVO2M. Supplementary Table S9: Estimated coefficients, standard errors (SE), p-values, partial $eta^2$, and Cohen's d. BMUSC. Supplementary Table S10: Estimated coefficients, standard errors (SE), p-values, partial $eta^2$, and Cohen's d. SJ. Supplementary Table S11: Estimated coefficients, standard errors (SE), p-values, partial $eta^2$, and Cohen's d. BFAT.

**Author Contributions:** Conceptualisation, A.F., A.I.F.G. and V.H.; methodology, R.B., B.L. and B.J.N.; formal analysis, R.B. and B.J.N.; investigation, V.H. and A.I.F.G.; resources, V.H. and A.F.; writing—original draft preparation, A.F., A.I.F.G., V.H., R.B. and B.L.; writing—review and editing, R.B., B.J.N., B.L., V.H. and A.F.; visualisation, R.B. and V.H. All authors have read and agreed to the published version of the manuscript.

**Funding:** This study was funded by the Slovenian Research Agency (grant numbers P5-0147).

**Institutional Review Board Statement:** The study was conducted in accordance with the Declaration of Helsinki and approved by the Ethics Committee of the University of Ljubljana, Faculty of Sport (2023-2).

**Informed Consent Statement:** Informed consent was obtained from all subjects involved in the study.

**Data Availability Statement:** The datasets used during the current study are available from the corresponding authors upon reasonable request.

**Conflicts of Interest:** The authors declare no conflict of interest.

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
