# Peer review of "COVID-19 and Changes in the Model of Physical Fitness and Body Composition of Young Tennis Players"

_applsci, doi:10.3390/app131810015_

Round 1

Reviewer 1 Report

Thanks for inviting me to review the article.

The quality of the article is good, but it is necessary to pay more attention to the sources of the first category in the introduction.

The necessity of conducting this study should be discussed more.

Comparing the results with previous studies in discussion and conclusions should be done better

The quality of writing the article is good

But it is better for a native reader to read it.

Author Response

Reviewer 1

Thanks for inviting me to review the article.

The quality of the article is good, but it is necessary to pay more attention to the sources of the first category in the introduction.

The necessity of conducting this study should be discussed more.

Comparing the results with previous studies in discussion and conclusions should be done better

The quality of writing the article is good

But it is better for a native reader to read it.

Author's answers to Reviewer 1 comments:

RESPONSE: Thank you very much for your suggestion. A native English speaker has checked the article. The changes are marked in blue in the article.

Reviewer 2 Report

I suggest to add separate, statistical analysis section as the part of Materials and Methods section.

Author Response

Reviewer 2

RESPONSE: Thank you for your suggestion. The Materials and Methods section has been divided into 3 subsections: 1) Research Design and Participants; 2) Data Collection; 3) Statistical Analysis.

A native English speaker has checked the article. The changes are marked in blue in the article.

Reviewer 3 Report

This retrospective study aimed to analyze the impact of the COVID-19 pandemic (2020-2021) on the physical fitness and body composition of young tennis players, in comparison to pre-pandemic trends (2015-2019). The study revealed that boys experienced an increase in body fat mass while observing a decrease in muscle mass during the pandemic. Surprisingly, age-adjusted squat jump test scores improved. Conversely, no significant differences were observed in other indicators for both boys and girls. The findings suggest that the prolonged reduction in sports activity due to the pandemic may have negatively affected athletes' body composition, although selected performance indicators remained unaffected.

Overall, the manuscript addresses an important topic and utilizes sound methods to investigate the impact of the COVID-19 pandemic on young tennis players' physical fitness and body composition. However, there are several areas that could benefit from improvement:

1. Methods:

Ensure that all variables are clearly defined, including their purpose and significance in the study. This will aid readers in better understanding the research design and interpretation of results.

2. Statistical Analysis:

It is commendable that the purpose of each data analysis strategy is indicated. However, consider providing explanations for the choice of analysis methods in the context of any limitations faced during the study. For instance, if certain analyses were chosen due to data constraints or other factors, these should be mentioned.

3. Figure 1:

Avoid repeating the sample size in each panel of the figure. Instead, state the total sample size for each year once, and it will be sufficient for readers to understand.

4. Figure Placement:

Consider moving Figure 3 to the Results section and Figure 4 to either the Results or Methods section, depending on the flow of the manuscript and the logical sequence of presenting findings and methods.

5. Discussion:

Ensure that the content in the Discussion section aligns well with the Results section. Some parts that primarily describe findings might be better placed in the Results section. The flow of the discussion should be improved to enhance clarity and coherence.

Overall, the study holds promise and contributes valuable insights into the effects of the pandemic on young tennis players' physical fitness and body composition. By addressing the mentioned areas, the manuscript will become more compelling and easier for readers to follow. Please consider these suggestions in your revisions, and I look forward to reviewing the updated version.

Author Response

Reviewer 3

This retrospective study aimed to analyze the impact of the COVID-19 pandemic (2020-2021) on the physical fitness and body composition of young tennis players, in comparison to pre-pandemic trends (2015-2019). The study revealed that boys experienced an increase in body fat mass while observing a decrease in muscle mass during the pandemic. Surprisingly, age-adjusted squat jump test scores improved. Conversely, no significant differences were observed in other indicators for both boys and girls. The findings suggest that the prolonged reduction in sports activity due to the pandemic may have negatively affected athletes' body composition, although selected performance indicators remained unaffected.

Overall, the manuscript addresses an important topic and utilizes sound methods to investigate the impact of the COVID-19 pandemic on young tennis players' physical fitness and body composition. However, there are several areas that could benefit from improvement:

  1. Methods:

Ensure that all variables are clearly defined, including their purpose and significance in the study. This will aid readers in better understanding the research design and interpretation of results.

RESPONSE: Thank you very much for your comment. We believe that all six variables that were included in the study were defined and abbreviated in a manner that is consistent with previous research. Given that they have also been well explained in several studies, we do not believe that additional descriptions of the variables is necessary. The changes relating to your comments/suggestions are marked throughout the article in red.

  1. Statistical Analysis:

It is commendable that the purpose of each data analysis strategy is indicated. However, consider providing explanations for the choice of analysis methods in the context of any limitations faced during the study. For instance, if certain analyses were chosen due to data constraints or other factors, these should be mentioned.

RESPONSE: We appreciate the positive feedback regarding our attempts to adequately describe the purpose of each step in the data analysis. Indeed, we strive for full transparency in how these analyses were carried out to ensure the process behind each individual step is clear. The choice of the statistical model, in particular its functional form (which we describe in detail in the manuscript), was in our case driven by the research question  using the data that we had available. The main limitation of our data, which greatly influenced our approach to the analysis method, was the considerable reduction in sample size for the 2020 measurements. We attempted to account for this limitation by stratifying by gender and adjusting for age. We have already outlined this detail in the original version of the manuscript, and thoroughly explained our reasoning in choosing this particular strategy. We are confident that the revisions we have made to the structure of the discussion section, as requested by Reviewer 2, have also helped to more clearly outline our rationale and subsequent approach.

  1. Figure 1:

Avoid repeating the sample size in each panel of the figure. Instead, state the total sample size for each year once, and it will be sufficient for readers to understand.

RESPONSE: Thank you very much for your suggestion. Based on your comment, we have changed the order of the figures. The sample size values are now listed in Figure 2. However, we must point out that there are slight variations in the sample values for individual variables, and it is of course very important for this to be outlined in all instances in which it could influence interpretation of data.

  1. Figure Placement:

Consider moving Figure 3 to the Results section and Figure 4 to either the Results or Methods section, depending on the flow of the manuscript and the logical sequence of presenting findings and methods.

RESPONSE: Thank you for your comment. We have changed the order of the figures. Figure 4, which shows the timing of suspensions, training periods and competitions, and annual testing, has been moved to the end of the Data Collection subsection (within the Materials and Methods section). We are somewhat hesitant to move Figure 3 (now Figure 4), and its explanation, to the Results or Methods section. The reason is that Figure 3 relates to an additional explanation as to why a surprising increase in squat jump height was observed in our sample. In our opinion, this is an important part of the article through which we suggest that the effects of interrupted or reduced physical activity may vary depending on individual physical characteristics and/or movement abilities.

  1. Discussion:

Ensure that the content in the Discussion section aligns well with the Results section. Some parts that primarily describe findings might be better placed in the Results section. The flow of the discussion should be improved to enhance clarity and coherence.

RESPONSE: We agree with your suggestion. We have shortened the discussion by moving some of the text to the Results section. In doing so, we believe the discussion section is now more concise and flows with a clearer narrative. All changes regarding this point are marked in red in the text.

Overall, the study holds promise and contributes valuable insights into the effects of the pandemic on young tennis players' physical fitness and body composition. By addressing the mentioned areas, the manuscript will become more compelling and easier for readers to follow. Please consider these suggestions in your revisions, and I look forward to reviewing the updated version.

RESPONSE: Thank you for reviewing the article and for your comments. We are grateful that the changes that we have made in response to your suggestions have made the article more transparent and readable.

A native English speaker has checked the article. The changes are marked in blue in the article.

Reviewer 4 Report

Please find it in the manuscript file.

The paper is well written. 

Author Response

Reviewer 4

RESPONSE: We thank you for the corrections you have made to the article. We have taken your suggestions fully into account. They are marked in green color.

A native English speaker has checked the article. The changes are marked in blue in the article.

Round 2

Reviewer 3 Report

Thank you very much for thoroughly addressing my comments. It is a great study. Thanks